# AudBility: Effectiveness of an online central auditory processing screening program

**Nádia Giulian de Carvalho**[1]*, **Maria Isabel Ramos do Amaral**[2], **Maria Francisca Colella-Santos**[3]

**1** Child and Adolescent Health Program—Pediatric Research Center-CIPED/FCM/UNICAMP, Campinas, Brazil, **2** Department of Human Development and Rehabilitation–DDHR/FCM/UNICAMP, Campinas, Brazil, **3** Department of Human Development and Rehabilitation and the Pediatric Research Center–DDHR/FCM/UNICAMP, Campinas, Brazil

* nadiagiulian@gmail.com

**Data Availability Statement:** All relevant data are within the paper.

**Funding:** The study was funded by FAPESP (2017/03317-6). FAPESP (fapesp.br/en) is a public foundation, funded by the taxpayer in the State of

## Abstract

### Objective

To contribute to the validation of AudBility, an online central auditory processing screening program, considering the tasks for age between 6 and 8 years-old, from the investigation of sensitivity and specificity, as well as to suggest a minimum central auditory processing (CAP) screening protocol in this age group.

### Method

In the first stage of the study, 154 schoolchildren were screened. Children were aged between 6 and 8 years old, native speakers of Brazilian Portuguese. The auditory tasks of AudBility analyzed in this study were: sound localization (SL), auditory closure (AC), figure-ground (FG), dichotic digits—binaural integration (DD), temporal resolution (TR) and temporal frequency ordering (TO-F). In the second stage, 112 children attended to CAP assessment in the institution's laboratory. The calculation of efficacy (sensitivity/specificity) was obtained through the construction of the ROC curve for the tests with more than five children altered in the diagnosis.

### Results

For the 6–7-year-old age group the accuracy values were: AC (76.9%); FG (61.6%); DD 78.8% for the right ear and 84.4% for the left ear in females and 63.2% for the left ear in males; TR (77.1%) and TO-F (74.4% for the right ear and 82.4% for the left ear). For the 8-year-old age group the values were: FF (76.5%); DD (71.7% for the left ear for females and 77% for the right ear for males); TR (56.5%) and TO-F (54.1% for the right ear and 70% for the left ear).

### Conclusions

AudBility showed variations in sensitivity and specificity values between the auditory tasks and age groups, with better effectiveness in schoolchildren between the ages of 6 and 7 than eight-year-olds, except for the FG task. For screening purposes, the application of the

São Paulo, with the mission to support research
projects in higher education and research
institutions, in all fields of knowledge.

**Competing interests:** The authors have declared
that no competing interests exist.

protocol involving five tasks for the 6 to 7-year-olds group and with four tasks for the 8-year-olds group is suggested.

## Introduction

Central Auditory Processing Disorder (CAPD) is considered a diagnostic entity, listed in ICD 10 as ear disease (code H93.25), which confirms the physiological nature of this disorder and the need for adequate support for schoolchildren [1]. It has been shown in the literature that schoolchildren with academic difficulties commonly present CAPD, suggesting delay in the maturation of auditory skills in this population [2, 3]. The absence of a single procedure that is the gold standard in the CAPD diagnosis, as well as a single conceptualization and universal criteria adopted for diagnosis reflect in rates that can vary from 7.3% to 96% depending on the criterion used [4, 5].

The behavioral characteristics often observed in children diagnosed with CAPD are difficulties in understanding speech in noisy environments or with competitive messages, in following a message presented quickly or with complex auditory commands, as well as difficulty in discriminating similar sounds [6, 7]. However, it is known that in view of the complexity of auditory mechanisms and heterogeneity of the disorder in the central auditory nervous system (CNS), such characteristics are not exclusive to CAP. Therefore, behavioral complaints obtained only through questionnaires and/or *checklists* are not sufficient in a screening program, and a battery of behavioral tests that access the auditory mechanisms/abilities recommended by American Speech-Language-Hearing Association (ASHA) is necessary [1, 8]. In addition, simply incorporating several behavioral tests into a battery does not guarantee its efficiency, and it is necessary to obtain sensitivity, specificity, and efficiency data from individual tests [6, 7].

Screening can be understood as the first step to access schoolchildren with potential risk of CAPD. The term screening, when associated with CAP, refers to the method that will determine the need (or not) for future tests for children who already exhibit some type of complaint/difficulty [9]. An effective screening method can promote awareness among educators and parents, contribute to a broader epidemiological survey of the school population, favor early intervention, and reduce costs of unnecessary referrals [10, 11].

Given this scenario, researchers have been looking since the 80s for the development of batteries/tests for the purpose of central auditory processing (CAP) screening in a school setting [12–15]. Currently, the development of more comprehensive batteries of auditory skills through interactive tools, such as the *Screening Test for Auditory Processing* (STAP) [16] and *Feather Squadron* [17] stand out in the international scenario. The use of technology in the health area has provided the requirements for screening procedures, considering the easy access and reduced application time, wide range of auditory mechanisms contemplated in the tasks, as well as playful and motivating activities for the child [14]. However, the need for sensitivity, specificity, reliability test and retest data to be studied and made available is also highlighted, contributing to the validation of new batteries for clinical use in school screening [18–20].

In Brazil, a screening battery called "AudBility" was developed. It is a computational system that includes auditory tasks such as sound localization (SL), auditory closure (AC), figure-ground (FG), dichotic digits—binaural integration (DD), temporal resolution (TR) and temporal frequency ordering (TO-F), accessible for children starting at 6 years-old. The first

version of AudBility was applied in schoolchildren with good school performance, in the 8- to 11-year-old age group. The authors observed the need for improvements and the elaboration of tasks for age between 6 and 8 years- old [21].

Thus, the objective of this study was to contribute to the validation of AudBility, considering the tasks for age between 6 and 8 years-old, from the investigation of sensitivity and specificity, as well as to suggest a minimum central auditory processing screening protocol (CAP) for this age group.

## Method

### Type, study location and ethical aspects

This is a diagnostic accuracy and cross-sectional study, with a prospective data collection.

The study was approved by the institution's Research Ethics Committee, under opinion No. 2.294.609. The students' parents/legal guardians gave consent for their voluntary participation in the research by signing the Free and Informed Consent Form (ICF) and the children also signed a Term of Consent.

The study was carried out in two stages. The first stage consisted in the screening of schoolchildren in a public-school setting. Subsequently, all those responsible for the screened children were invited to attend the audiology Laboratory of Institution for the second stage of the study within a 72-hour period, where the CAP screening was carried out. Both stages were performed by one of the signatories of the study, an audiologist with experience in the area and familiar with the collection procedures.

## Stage 1—Central auditory processing screening in a school setting

### Sample

This is a convenience sample from a Public School in the city of XXXXX, Brazil. Two hundred and three invitation letters were sent to parents/legal guardians of students enrolled in the 1$^{st}$, 2nd, and 3rd grade of elementary school and 157 (77%) of them agreed to participate in the research.

**Inclusion criteria.** Age group between 6 and 8 years old, native speakers of Brazilian Portuguese, normal peripheral screening procedures, child without previous diagnosis of syndromes, cognitive or neurodevelopmental disorders.

**Exclusion criterion.** Children who did not have adequate understanding during the screening battery tasks.

Three children were excluded from CAP screening due to difficulty understanding the auditory tasks. The schoolchildren were referred for medical evaluation. Based on the inclusion/exclusion criteria, the sample at this stage consisted of 154 schoolchildren.

### Procedures

The auditory screening was carried out individually in a computer room provided by the school. Firstly, peripheral hearing screening was performed, consisting of otoscopy (Hein otoscope) and immittance measures (MT-10 Interacoustics equipment), including tympanometry and ipsilateral acoustic reflex at 500Hz, 1KHz, 2Kz and 4KHz. Children who passed this screening were then submitted to the CAP screening, through the AudBility—tasks for age between 6–8 years old.

A desktop computer provided by the school was used to access the AudBility online platform. The computer's volume mixer was set at 50% and the child used a noise-canceling headset, a technology that reduces noise by 95% (26dB) (Panasonic supra-headset model:

RPHC720). For the sensitivity and specificity study, the auditory tasks of sound localization (SL), temporal resolution (TR), auditory closure (AC), dichotic digits—binaural integration (DD), figure-ground (FG) for verbal sounds and temporal frequency ordering (TO-F) were analyzed. The children's answers for each task were selected on the screen by the audiologist. The descriptions of the auditory tasks were previously reported [21] and are summarized in Table 1.

## Stage 2—Central auditory processing (CAP) assessment

In the second stage of the study, 112 children attended to Laboratory of Institution, being 61 children in the 6- to 7-year-old age group and 51 from the 8-year-old age group. Initially, case history was carried out with the parents/legal guardians of the children. Subsequently, a basic audiological evaluation as well as a CAP behavioral assessment were carried out. The evaluations were performed in an acoustic booth, using a duly calibrated AC40- Interacoustics audiometer and TDH 49 headphones and a Dell brand notebook.

For each task of the AudBility Program, an equivalent test was applied to evaluate the same auditory skills screened in the school environment.

The CAP behavioral assessment battery protocol consisted of the following tests:

1. Masking Level Difference (MLD) [22].

2. Random Gap Detection Test (RGDT) [23].

3. Frequency Patterns Test (FPT) [24].

4. Dichotic Digit Test (DD) in the binaural integration stage [25].

5. Speech in noise test (FR) [25].

6. Pediatric Speech Intelligibility with ipsilateral competitive message (PSI) in the signal-to-noise ratio -15 [25].

**Table 1. Auditory tasks from the AudBility screening for age between 6–8 years old.**

| Auditory tasks | Brief description of the hearing tasks |
|---|---|
| 1. Sound localization (SL) | 10 sequences in which the child hears sounds that represent everyday activities. The child must indicate the correct direction with respect to the location of the target stimulus (right, left, back/up or right and left). |
| 2. Auditory Figure-Ground (FG) | 10 sequences per ear in which the child hears a story (noise) and concomitantly a phrase (sign) referring to the figure and must point to the figure. Five sentences are presented in the signal-to-noise ratio of -10 dB, and five sentences are presented in the signal-to-noise ratio of -15 dB. |
| 3. Dichotic digits binaural integration (DD) | 10 presentations with 4 numbers presented dichotically (two in the right ear and two in the left ear). The screen will always show all numbers from 1 to 9 as options so that the four digits heard can be chosen. |
| 4. Auditory closure (AC) | 10 sequences per ear: the child hears a word modified acoustically using the Gargle effect option of the EarMix software (CTS Informática) and must recognize the word among the figures shown on the screen. |
| 5. Temporal resolution (TR) | 10 sequences with 1000Hz tone with intervals between them—the gaps—which will have variations of 20ms, 15ms, 10ms, 6ms, 4ms and 0ms presented at random. In each presentation, the child hears a sequence of six sounds, both single and double, and is told to count how many double sounds he/she can perceive/hear. |
| 6. Temporal Ordering Frequency (TO-F) | 5 sequences of three tones per ear: bass (GROSSO-G) of 700Hz and treble (FINO-F) of 1500Hz. The inter-stimulus interval is 350ms. The child must hear and name the correct sequence (GGF, FFG, FGF, GFG, GFF and FGG). |

The established criterion for the diagnosis of CAPD was below-normal performance in at least two tests [6].

## Analysis

The ROC (*Receiver Operating Characteristic*) curve is a graphical representation that requires an expressive number of individuals in the two sets that need to be differentiated (in this case, "normal" and "altered"). Therefore, tests with less than five individuals altered could not have their values calculated. The value the sensitivity, specificity and efficiency [26] and coefficient J [27], considered the best equilibrium point between sensitivity and specificity was analyzed. The area under the curve (AUC) value indicates that an ear with an altered diagnostic result has this probability (AUC x 100%) of performing worse in screening.

From the CAP perspective, sensitivity can be understood as the ability of the test to correctly identify individuals who have CAPD (true positive rate) and specificity is the ability to correctly identify normal individuals (true negative rate). The relationship between sensitivity and specificity (area value under the ROC curve) represents the accuracy of the test since it considers all sensitivity and specificity values for each value of the test variable [28].

In a preliminary study, under review, conducted with a sample of schoolchildren with good school performance, statistically significant differences ($p \leq 0.05$) were found between the ages of 6–7 years old compared to 8 years old and better performance of males in the DD task for the right ear as well for the right ear in the DD and OT-F tasks. Therefore, these variables were considered for the study of sensitivity/specificity.

## Results

Table 2 presents the distribution of the individuals who took part in the CAP clinical tests.

Figs 1 and 2 show the ROC curves for the screening tasks in the 6 to 7- and 8-year-olds age group, respectively.

Table 3 shows the values of sensitivity, specificity, efficiency, and coefficient J, for the screening tasks for schoolchildren in the 6- to 7-year-old age group.

Table 4 shows the values of sensitivity, specificity, efficiency, and coefficient J, for schoolchildren in the 8-year-old age group.

Table 5 shows a protocol suggestion based on the accuracy of the screening, obtained through the result between sensitivity and specificity (area value under the ROC curve).

## Discussion

Screening auditory skills and identifying schoolchildren in the literacy phase with possible risk of CAPD is an advance in audiology because these children may face many challenges in the school environment and the sooner the child is referred for differential diagnosis, the sooner they can receive the appropriate interventions.

In this study, the effectiveness rate of six auditory tasks was studied through sensitivity and specificity. It is noteworthy that in none of the age groups studied, it was possible to predict the sensitivity and specificity of the sound localization task (SL) of AudBility through the construction of the ROC curve (Table 2). The MLD, a diagnostic test corresponding to the SL screening task, evaluates the lower region of the brain stem, with expected maturation in the first years of life. Therefore, it is not surprising that from the age of 6 the performance became easy, with few changes, since the age of five years has already been referred to in the literature as a period to reach the level of performance in adults [29].

The value of accuracy in the auditory closure (AF) screening task could be obtained in the 6- to 7-year-old age group, but it could not be obtained in the 8-year-olds age group,

**Table 2. Performance of schoolchildren in the central auditory processing behavioral assessment.**

| Test | Age group | Results | Absolute frequency (n) | Relative frequency (%) |
|---|---|---|---|---|
| MLD | 6–7 years-old | Normal | 60 | 98.36 |
| | | Altered | **1** | 1.64 |
| | 8 years old | Normal | 48 | 94.12 |
| | | Altered | **3** | 5.88 |
| FR | 6–7 years-old | Normal | 116 | 95.08 |
| | | Altered | 6 | 4.92 |
| | 8 years old | Normal | 101 | 99.02 |
| | | Altered | **1** | 0.98 |
| RGDT | 6–7 years-old | Normal | 47 | 77.05 |
| | | Altered | 14 | 22.95 |
| | 8 years old | Normal | 46 | 90.20 |
| | | Altered | 5 | 9.80 |
| DD-RE - ♀ | 6–7 years-old | Normal | 19 | 55.88 |
| | | Altered | 15 | 44.12 |
| | 8 years old | Normal | 19 | 82.61 |
| | | Altered | **4** | 17.39 |
| DD-RE - ♂ | 6–7 years-old | Normal | 23 | 85.19 |
| | | Altered | **4** | 14.81 |
| | 8 years old | Normal | 23 | 82.14 |
| | | Altered | 5 | 17.86 |
| DD-LE - ♀ | 6–7-year-olds | Normal | 19 | 55.88 |
| | | Altered | 15 | 44.12 |
| | 8 years old | Normal | 18 | 78.26 |
| | | Altered | 5 | 21.74 |
| DD-LE - ♂ | 6–7 years-old | Normal | 22 | 81.48 |
| | | Altered | 5 | 18.52 |
| | 8 years old | Normal | 24 | 85.71 |
| | | Altered | **4** | 14.29 |
| PSI | 6–7 years-old | Normal | 103 | 84.43 |
| | | Altered | 19 | 15.57 |
| | 8 years old | Normal | 95 | 95.00 |
| | | Altered | 5 | 5.00 |
| FPT-RE | 6–7 years-old | Normal | 18 | 29.51 |
| | | Altered | 43 | 70.49 |
| | 8 years old | Normal | 38 | 74.51 |
| | | Altered | 13 | 25.49 |
| FPT-LE | 6–7 years-old | Normal | 14 | 22.95 |
| | | Altered | 47 | 77.05 |
| | 8 years old | Normal | 30 | 58.82 |
| | | Altered | 21 | 41.18 |

demonstrating that the speech in noise diagnostic test is easy to perform for older children. In the present study, it is relevant to highlight the accuracy value of 76.9% obtained for children in the 6- to 7-year-old age group, through the ROC curve, with sensitivity and specificity values of 66.67% and 73.28%, respectively. It is noteworthy that they are high values that confirm the validity of the AudBility task in screening the AC ability. This ability is important for school-children in the early years of the literacy process, and it is known that difficulty in speech

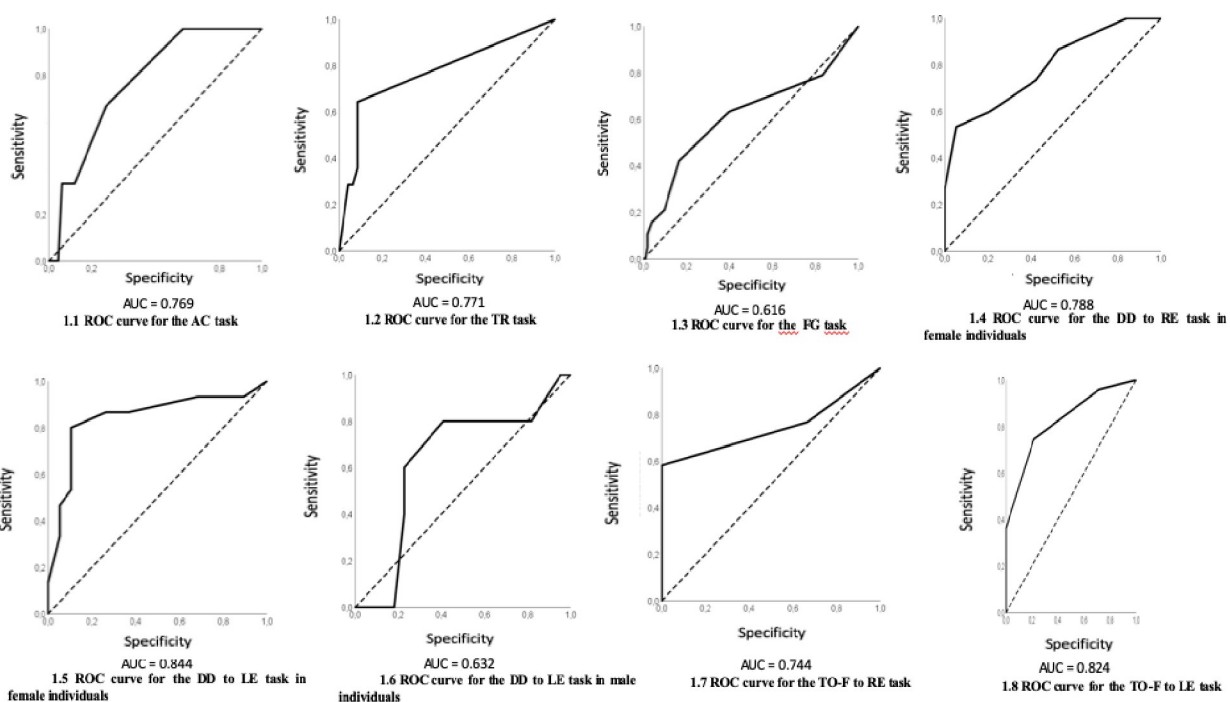

**Fig 1. ROC curves for the auditory tasks of schoolchildren in the between 6- to 7-year-old age group.**

perception in noise can result in poor phonological representations, reading difficulties, memory and learning difficulties [30, 31]. In a previous study, authors found lower values of accuracy and sensitivity obtained through the filtered speech diagnostic test, applied in individuals

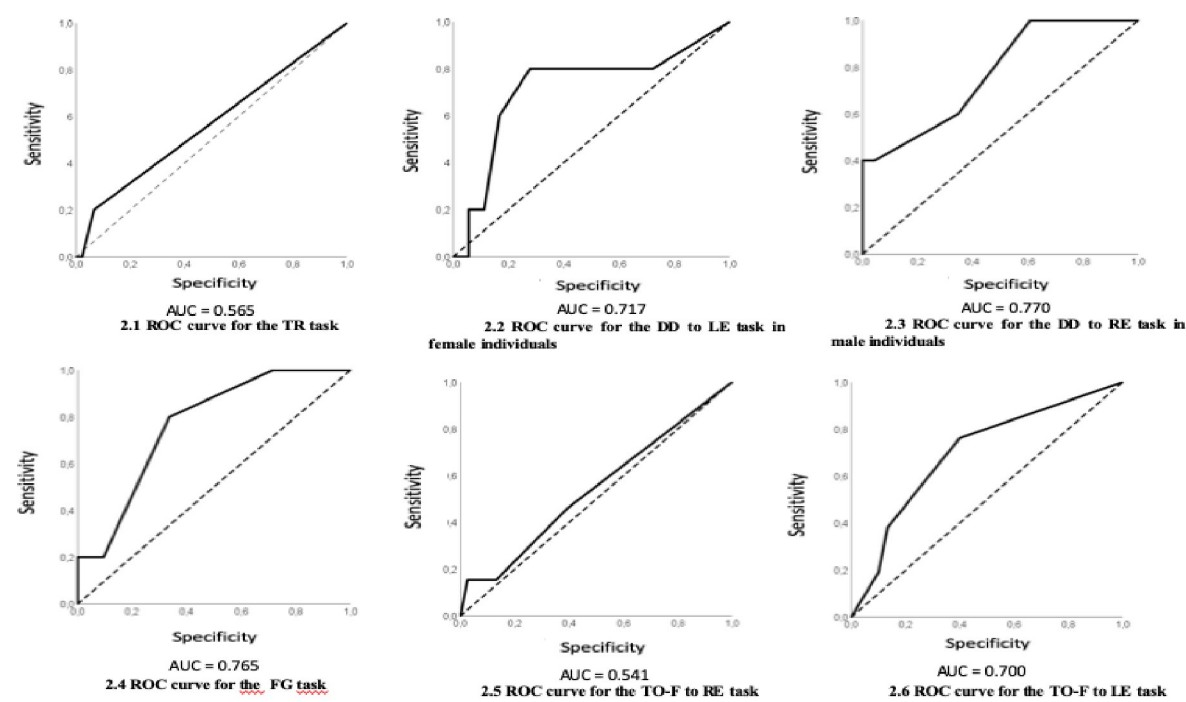

**Fig 2. ROC curves for the auditory tasks of schoolchildren in the 8-year-old age group.**

**Table 3. Performance of schoolchildren in the 6- to 7-year-old age group in AudBility and values of sensitivity, specificity, and efficiency.**

| Auditory tasks | Sensitivity (%) | Specificity (%) | False positive (%) | False negative (%) | Efficiency (%) | J |
|---|---|---|---|---|---|---|
| Auditory Closure | 66.67 | 73.28 | 26.72 | 33.33 | 69.97 | 0.399 |
| Temporal Resolution | 64.29 | 91.49 | 8.51 | 35.71 | 77.89 | 0.558 |
| Figure-Ground | 42.11 | 83.50 | 16.50 | 57.89 | 62.80 | 0.256 |
| DD-Binaural integration (RE) females | 53.33 | 94.74 | 5.26 | 46.67 | 74.04 | 0.481 |
| DD-Binaural integration (LE) females | 80.00 | 89.47 | 10.53 | 20.00 | 84.74 | 0.695 |
| DD-Binaural integration (LE) males | 80.00 | 59.09 | 40.91 | 20.00 | 69.55 | 0.391 |
| Temporal Frequency Ordering RE | 58.14 | 100.00 | 0.00 | 41.86 | 79.07 | 0.581 |
| Temporal Frequency Ordering LE | 74.47 | 78.57 | 21.43 | 25.53 | 76.52 | 0.530 |

between 13–59 years old who had lesions in the central auditory system, with a value of 65%, which resulted in sensitivity of 25% and 91% specificity [19].

In the temporal resolution screening task (TR), an accuracy value of 77.1% was obtained in children in the 6- to 7-year-old age group and 56.5% at the 8-year-old age group. Therefore, the contribution of this screening task, especially when performed by younger children, is observed because the TR ability may influence important aspects related to the initial literacy process, such as the perception of consonant groups in speech and writing, perception of tonic syllable and deaf and sound phonemes.

The figure-ground screening (FG) task was the only one in which 8-year-olds had a higher accuracy value (76.5%) compared to the 6 to 7-year-old group (61.6%), being an important screening task, since the FG ability contributes to the adequate understanding of speech in competitive environments and school tasks and children with academic difficulties may present worse speech perception in noise [31].

In the dichotic digit screening task—binaural integration (DD) the obtained accuracy values diverged between ears and gender. The performance was better for the females in the 6 to 7-year-old age group for the left ear (LE) (84.4%). In the 8-year-old age group, it was only possible to establish the performance of the LE for the females and the RE for the males. In a previous study aimed at assessing sensitivity/ specificity in the dichotic digits screening task performed by individuals with lesions, the authors also found high accuracy in the test results, being 90%, with 80% sensitivity and 89% specificity [19]. It is noteworthy that dichotic listening tests are sensitive to evaluate cortical structures and mechanisms of binaural integration, in addition, they have a strong correlation with learning difficulties [32], therefore, it has been referred to in the literature as a potential test to integrate a screening program [33].

In the temporal frequency ordering (TO-F) screening task the values obtained for accuracy diverged between ears, with better results for children in the 6 to 7-year-old group (74.4% - 82.4%). It is noteworthy that the TO-F task is particularly important in a screening protocol,

**Table 4. Performance of 8-year-old schoolchildren in AudBility and values of sensitivity, specificity, and efficiency.**

| Auditory tasks | Sensitivity (%) | Specificity (%) | False positive (%) | False negative (%) | Efficiency (%) | J |
|---|---|---|---|---|---|---|
| Temporal Resolution | 20.00 | 93.48 | 6.52 | 80.00 | 56.74 | 0.135 |
| Figure-Ground | 80.00 | 66.32 | 33.68 | 20.00 | 73.16 | 0.463 |
| DD-Binaural integration (LE) females | 80.00 | 72.22 | 27.78 | 20.00 | 76.11 | 0.522 |
| DD-Binaural integration (RE) males | 40.00 | 100.00 | 0.00 | 60.00 | 70.00 | 0.400 |
| Temporal Frequency Ordering RE | 15.38 | 97.37 | 2.63 | 84.62 | 56.38 | 0.128 |
| Temporal Frequency Ordering LE | 76.19 | 60.00 | 40.00 | 23.81 | 68.10 | 0.362 |

**Table 5. CAP screening protocol recommended for schoolchildren in the 6- to 8-year-old age group.**

| Auditory tasks | Accuracy (%) 6–7 years-old | Accuracy (%) 8 years old |
|---|---|---|
| Auditory Closure | 76.9 | - |
| Figure-Ground | 61.6 | 76.5 |
| Dichotic digits | 63.2–84.4 | 71.7–77.0 |
| Temporal resolution | 77.1 | 56.5 |
| Temporal ordering | 74.4–82.4 | 54.1–70.0 |

because often children with a CAPD diagnosis and school difficulties score below normal limits in the frequency patterns diagnostic test [34, 35]. In a previous study, which analyzed the results of the temporal ordering of frequency screening test in individuals with lesions in the central auditory system, the authors found accuracy of 68%, which resulted in sensitivity of 83% and specificity of 100% [19].

We observed that in most auditory tasks the accuracy value results were higher for children in the 6- to 7-year-old age group, except for the figure-ground ability. In the diagnostic process, the use of several tests can potentially reduce the error, improving the efficiency of the battery, since it encompasses a greater range of auditory mechanisms and provides more appropriate information for the establishment of conducts, and it is recommended that a central auditory processing diagnostic battery have two or more tests with established reliability, validity, sensitivity, specificity, and efficiency [19]. Therefore, following the same principles, it is suggested that the screening protocol for children between 6 and 7 years old be applied with five tasks (auditory closure, figure-ground, binaural integration, resolution and temporal ordering) and at 8 years old with four tasks (figure-ground, binaural integration, resolution and temporal ordering), since in these tasks it was possible to contribute to the validation of the data. It is noteworthy that both protocols recommend the inclusion of at least one dichotic listening test, a monaural test of low redundancy and a temporal ordering test in a screening battery program [10].

Given the results found in this study, we suggest the use of AudBility in screening programs, also considering the complaints associated with the results obtained in the battery for decision making for referral. A recent study reported that the central auditory processing screening battery, called STAP, had validated sensitivity and specificity values of 76.6% and 72%, respectively. These values improved when the use of the questionnaire (SCAP) was added to the behavioral tests [16]. The sensitivity and specificity values found in this study demonstrated that AudBility contains tasks with high efficacy values. The specialized literature has already reported that CAP screening batteries applied in children had low sensitivity, being reported sensitivity of 30–40% in the *Multiple Auditory Processing Assessment (*MAPA) and 45% in the SCAN batteries [15].

In this study, the feasibility of the application was observed with children starting at 6 years old, corroborating with a study that demonstrated the feasibility of screening with 93% of children starting at five years old [17]. A recent longitudinal study demonstrated that children with poor auditory perception in a CAP battery at five years old similarly presented a lower score at seven years old [36]. AudBility should not be a battery that replaces the need for a thorough CAP behavioral-auditory diagnostic evaluation. This is the first validation study considering the tasks for age between aimed at children between 6 and 8 years-old, therefore new research should be carried out.

## Conclusion

AudBility showed variations in sensitivity and specificity values between auditory tasks and age groups, with better effectiveness in schoolchildren aged between 6 and 7 years than eight-year-olds, except for the FG task. For screening purposes, the application of the protocol involving five tasks for the 6–7 age group and with four tasks for the 8-year-olds group is suggested.

## Limitations

The study has limitations in relation the was not application of formal tests to assess language and cognitive aspects. In addition, peripheral cochlear screening was not performed.

## Acknowledgments

We thank the speech therapists Dr. Ingrid Gielow and Msc. Diana Faria for the idealization of AudBility and the "E.E. Dona Castorina Cavalheiro" school staff.

## Author Contributions

**Conceptualization:** Nádia Giulian de Carvalho, Maria Isabel Ramos do Amaral, Maria Francisca Colella-Santos.

**Data curation:** Nádia Giulian de Carvalho, Maria Isabel Ramos do Amaral, Maria Francisca Colella-Santos.

**Funding acquisition:** Nádia Giulian de Carvalho.

**Investigation:** Nádia Giulian de Carvalho, Maria Isabel Ramos do Amaral.

**Methodology:** Nádia Giulian de Carvalho, Maria Isabel Ramos do Amaral, Maria Francisca Colella-Santos.

**Project administration:** Nádia Giulian de Carvalho, Maria Isabel Ramos do Amaral.

**Supervision:** Maria Isabel Ramos do Amaral, Maria Francisca Colella-Santos.

**Visualization:** Maria Francisca Colella-Santos.

**Writing – original draft:** Nádia Giulian de Carvalho.

**Writing – review & editing:** Nádia Giulian de Carvalho, Maria Isabel Ramos do Amaral, Maria Francisca Colella-Santos.

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
