## [Decision Letter · Decision Letter 0]

19 Mar 2021

PONE-D-21-02111

AudBility: effectiveness of an online central auditory processing screening program

PLOS ONE

Dear Dr. Carvalho,

Thank you for submitting your manuscript to PLOS ONE. After careful consideration, we feel that it has merit but does not fully meet PLOS ONE’s publication criteria as it currently stands. Therefore, we invite you to submit a revised version of the manuscript that addresses the points raised during the review process.

We look forward to receiving your revised manuscript.

Kind regards,

Qian-Jie Fu, Ph.D.

Academic Editor

PLOS ONE

Journal Requirements:

Additional Editor Comments (if provided):

Reviewers' comments:

Reviewer's Responses to Questions

**Comments to the Author**

1. Is the manuscript technically sound, and do the data support the conclusions?

Reviewer #1: Yes

Reviewer #2: Partly

2. Has the statistical analysis been performed appropriately and rigorously? 

Reviewer #1: Yes

Reviewer #2: Yes

3. Have the authors made all data underlying the findings in their manuscript fully available?

Reviewer #1: Yes

Reviewer #2: Yes

4. Is the manuscript presented in an intelligible fashion and written in standard English?

Reviewer #1: Yes

Reviewer #2: Yes

5. Review Comments to the Author

Reviewer #1: In Methods the authors state something about a "second stage." But I do not see any further discussion of a second stage, so what does that mean?

Line 32 The authors use the term "module" but I think that is the wrong word and they mean to discuss "the age range of subjects is....."

Line 54 The authors discuss "better than the eight year olds." Better is a relative term, better than what? I think the authors need to reword that sentence.

Line 172 or so the word "WAS" appears, I think that is an error

Line 183 Add the word and "...and better..."

Line 194 The authors need to use larger fonts for the figure legends. The font they use is difficult to read. Perhaps the journal redactor would have done that - but it needs to be changed.

Line 253 I made a comment but cannot read my own handwriting. That's embarrassing. But the comment has to do with transition to adults 29?

Line 325 The text should read something like: "...with better effectiveness than...eight year olds."

Reviewer #2: Thank you for sending me this manuscript. Firstly, The seccion of the limitation must be add. Every single publication have the limitations so it is crucial to add this secction. Then the checking of the manuscript will be possible.

6. PLOS authors have the option to publish the peer review history of their article (what does this mean?). If published, this will include your full peer review and any attached files.

Reviewer #1: **Yes: **Robert W. Keith, Ph.D.

Reviewer #2: No

---

## [Author Response · Author response to Decision Letter 0]

6 May 2021

Answer to the reviewers

First and foremost, we would like to thank you for your comments and suggestions. They were taken into account and have been implemented in our manuscript, which made it more appropriate and more interesting for publication. Therefore, please accept this new revised version for analysis.

Reviewer #1: In Methods the authors state something about a "second stage." But I do not see any further discussion of a second stage, so what does that mean?

Answer: This means that the study was applied in two stages, in two different locations. In the first stage, the screening was carried out at a school. In the second stage, students were invited to attend Laboratory of Institution for formal assessment of central auditory processing within a 72-hour period. In this second stage, we rely on the voluntary displacement of children and their families.

Line 32 The authors use the term "module" but I think that is the wrong word and they mean to discuss "the age range of subjects is....."

Answer: Thanks for the correction. Initially we thought of a module because AudBility was divided with tasks designed for children between 6 and 8 years old and above 9 years old. We made the correction and highlighted it in the text. 

Line 54 The authors discuss "better than the eight year olds." Better is a relative term, better than what? I think the authors need to reword that sentence.

Answer: Thanks for the correction. The term better is together with performance, results or effectiveness.

Line 172 or so the word "WAS" appears, I think that is an error

Answer: Thanks for the correction. This word has been deleted. 

Line 183 Add the word and "...and better..."

Answer: Thanks for the correction. The word and has been added.

Line 194 The authors need to use larger fonts for the figure legends. The font they use is difficult to read. Perhaps the journal redactor would have done that - but it needs to be changed.

Answer: Thanks for the correction. The font has been altered.

Line 253 I made a comment but cannot read my own handwriting. That's embarrassing. But the comment has to do with transition to adults 29?

Answer: The cited literature [29] has already reported that “Children of 5 yr of age did not perform significantly different from adults on the sound localization task”. Therefore, it is not surprising that from the age of 6 the performance became easy.

Line 325 The text should read something like: "...with better effectiveness than...eight year olds."

Answer: Thanks for the correction. 

Reviewer #2: Thank you for sending me this manuscript. Firstly, The seccion of the limitation must be add. Every single publication have the limitations so it is crucial to add this secction. Then the checking of the manuscript will be possible.

Answer: We appreciate and agree with your suggestion. The limitations section was added. Please check in the text.

---

## [Decision Letter · Decision Letter 1]

22 Jun 2021

PONE-D-21-02111R1

AudBility: effectiveness of an online central auditory processing screening program

PLOS ONE

Dear Dr. Carvalho,

Thank you for submitting your manuscript to PLOS ONE. After careful consideration, we feel that it has merit but does not fully meet PLOS ONE’s publication criteria as it currently stands. Therefore, we invite you to submit a revised version of the manuscript that addresses the points raised during the review process.

We look forward to receiving your revised manuscript.

Kind regards,

Qian-Jie Fu, Ph.D.

Academic Editor

PLOS ONE

Journal Requirements:

Reviewers' comments:

Reviewer's Responses to Questions

**Comments to the Author**

1. If the authors have adequately addressed your comments raised in a previous round of review and you feel that this manuscript is now acceptable for publication, you may indicate that here to bypass the “Comments to the Author” section, enter your conflict of interest statement in the “Confidential to Editor” section, and submit your "Accept" recommendation.

Reviewer #1: (No Response)

Reviewer #2: (No Response)

2. Is the manuscript technically sound, and do the data support the conclusions?

Reviewer #1: Yes

Reviewer #2: Yes

3. Has the statistical analysis been performed appropriately and rigorously? 

Reviewer #1: No

Reviewer #2: Yes

4. Have the authors made all data underlying the findings in their manuscript fully available?

Reviewer #1: Yes

Reviewer #2: Yes

5. Is the manuscript presented in an intelligible fashion and written in standard English?

Reviewer #1: Yes

Reviewer #2: Yes

6. Review Comments to the Author

Reviewer #1: 275 The value of accuracy in the auditory closure (AF) screening task could (note that AF is wrong.

Line 275 should read, “….. in the auditory closure (AC) screening….

In the summary spell out AD, FG and DD. The abbreviations do not work here, since some readers will go to the summery to decide whether to read the entire article. For example:

I think that AD that is shown above as copied from the manuscript is not correct - that the line should read auditory closure as AD and not AD

275 The value of accuracy in the auditory closure (AF) screening task

accuracy values were: AC (76.9%); FG (61.6%); DD 78.8%

dichotic digit screening task - binaural integration (DD)

figure-ground screening (FG) task w\\

Revise the line 354 where the English on limitations is not quite correct from:

354 The study has limitations in relation the was not application of formal tests to assess language and cognitive aspects. In addition, peripheral cochlear screening was not performed.

Reviewer #2: Thank you for giving me the opportunity to revise the manuscript “effectiveness of an online central auditory processing screening program” for publication in theJournal. I appreciate the time and effort that authors dedicated to prepare and conduct this study. Firstly, It would be great if linguistic proofreading will be done.

My comments:

L 54: change eight into 8

Could you add table were according to the PICO classification (Population with inclusion and exclusion criteria, I - intervention, C - comparator, o- outcome) you will explain each point?

The limitations of this study should be extended and the risk of this limitation should be add at the discussion.

7. PLOS authors have the option to publish the peer review history of their article (what does this mean?). If published, this will include your full peer review and any attached files.

Reviewer #1: **Yes: **Robert W. Keith

Reviewer #2: No

---

## [Author Response · Author response to Decision Letter 1]

5 Aug 2021

First and foremost, we would like to thank you for new comments and suggestions. They were taken into account and have been implemented in our manuscript. Therefore, please accept again this version with minor revision.

Reviewer #1: 275 The value of accuracy in the auditory closure (AF) screening task could (note that AF is wrong. Line 275 should read, “….. in the auditory closure (AC) screening….

Answer: There was a typo. We appreciate the correction.

In the summary spell out AD, FG and DD. The abbreviations do not work here, since some readers will go to the summery to decide whether to read the entire article. For example: I think that AD that is shown above as copied from the manuscript is not correct - that the line should read auditory closure as AD and not AD 275. The value of accuracy in the auditory closure (AF) screening task accuracy values were AC (76.9%); FG (61.6%); DD 78.8% dichotic digit screening task -binaural integration (DD) figure-ground screening (FG) task.

Answer: Abbreviations have been replaced by task names.

Revise the line 354 where the English on limitations is not quite correct from:

354 The study has limitations in relation the was not application of formal tests to assess language and cognitive aspects. In addition, peripheral cochlear screening was not performed.

Answer: The English has been revised. 

Reviewer #2: Thank you for giving me the opportunity to revise the manuscript “effectiveness of an online central auditory processing screening program” for publication in the Journal. I appreciate the time and effort that authors dedicated to prepare and conduct this study. Firstly, It would be great if linguistic proofreading will be done. My comments: L54: change eight into 8

Could you add table were according to the PICO classification (Population with inclusion and exclusion criteria, I - intervention, C - comparator, o- outcome) you will explain each point? The limitations of this study should be extended and the risk of this limitation should be add at the discussion.

Answer: We appreciate the comments. Changes were implemented in the manuscript. The linguistic proofreading was done; has been changed for eight. The PICO classification was added, and limitations has been extended.

---

## [Editor Report · Decision Letter 2]

11 Aug 2021

AudBility: effectiveness of an online central auditory processing screening program

PONE-D-21-02111R2

Dear Dr. Carvalho,

We’re pleased to inform you that your manuscript has been judged scientifically suitable for publication and will be formally accepted for publication once it meets all outstanding technical requirements.

Kind regards,

Qian-Jie Fu, Ph.D.

Academic Editor

PLOS ONE
---

## [Editor Report · Acceptance letter]

16 Aug 2021

PONE-D-21-02111R2 

AudBility: effectiveness of an online central auditory processing screening program 

Dear Dr. de Carvalho:

I'm pleased to inform you that your manuscript has been deemed suitable for publication in PLOS ONE. Congratulations! Your manuscript is now with our production department. 

Kind regards, 

on behalf of

Dr. Qian-Jie Fu 

Academic Editor

PLOS ONE